 

# Differential requirement of kindlin-3 for T cell progenitor homing to the non-vascularized and vascularized thymus

**Federico Andrea Moretti[1†], Sarah Klapproth[1†], Raphael Ruppert[1], Andreas Margraf[2], Jasmin Weber[2], Robert Pick[2], Christoph Scheiermann[2], Markus Sperandio[2], Reinhard Fässler[1], Markus Moser[1*]**

[1]Department Molecular Medicine, Max-Planck-Institute of Biochemistry, Martinsried, Germany; [2]Walter Brendel Center of Experimental Medicine, Biomedical Center, Ludwig-Maximilians-Universität, Martinsried, Germany

**Abstract** The role of integrin-mediated adhesion during T cell progenitor homing to and differentiation within the thymus is ill-defined, mainly due to functional overlap. To circumvent compensation, we disrupted the hematopoietic integrin regulator kindlin-3 in mice and found a progressive thymus atrophy that is primarily caused by an impaired homing capacity of T cell progenitors to the vascularized thymus. Notably, the low shear flow conditions in the vascular system at midgestation allow kindlin-3-deficient fetal liver-derived T cell progenitors to extravasate via pharyngeal vessels and colonize the avascular thymus primordium. Once in the thymus, kindlin-3 promotes intrathymic T cell proliferation by facilitating the integrin-dependent crosstalk with thymic antigen presenting cells, while intrathymic T cell migration, maturation into single positive CD4 and CD8 T cells and release into the circulation proceed without kindlin-3. Thus, kindlin-3 is dispensable for integrin-mediated T cell progenitor adhesion and signalling at low and indispensable at high shear forces.
DOI: https://doi.org/10.7554/eLife.35816.001

**\*For correspondence:**
moser@biochem.mpg.de

[†]These authors contributed equally to this work

## Introduction

T cell progenitors develop in the fetal liver (FL) or in the bone marrow (BM) and require the specialized environment of the thymus to mature into single positive (SP) CD4 or CD8 T cells (*Cai et al., 2007*; *Foss et al., 2001*; *Jotereau et al., 1987*). FL-derived T cell progenitors start colonizing the thymus anlage at around embryonic day (E) 11.5. Since the thymus anlage is initially avascular, T cell progenitors exit the adjacent pharyngeal vessels, migrate towards the thymic primordium, cross the dense extracellular matrix of the thymic capsule and finally settle in the thymic tissue. After thymus vascularization, which occurs at around E15 (*Liu et al., 2006*), T cell progenitors enter the thymus by extravasating from post-capillary venules at the cortico-medullary junction. Once in the thymus the progenitor cells interact and receive signals from thymic epithelial cells, macrophages, dendritic cells (DCs) and extracellular matrix proteins, which are essential for their proliferation, differentiation, selection and release into the circulation as mature, single positive T cells (*Boehm and Bleul, 2006*; *Koch and Radtke, 2011*).

Leukocyte extravasation follows a cascade of adhesion events commencing with selectin-mediated rolling of leukocytes on endothelial cells followed by the interaction with endothelial cell-bound chemokines, integrin-mediated leukocyte arrest and crawling and finally the transmigration through the vessel wall (*Ley et al., 2007*). Although integrins are expressed on all hematopoietic cells, conflicting reports exist on their role for T cell progenitor extravasation. While the treatment with function-blocking antibodies against α4 and/or β2 integrins reduced the homing of thymic progenitors

to the adult thymus (*Rossi et al., 2005*; *Scimone et al., 2006*), genetic ablation of neither α4 (*Arroyo et al., 1996*), nor αL (*Schmits et al., 1996*), α6 (*Georges-Labouesse et al., 1996*), β2 (*Scharffetter-Kochanek et al., 1998*), β7 integrins (*Bungartz et al., 2006*; *Wagner et al., 1996*), or the integrin ligands intercellular adhesion molecule (ICAM)−1 and vascular cell adhesion molecule (VCAM)−1 (*Koni et al., 2001*; *Sligh et al., 1993*; *Xu et al., 1994*) interfered with thymic colonization. Similarly, it is not known whether migration of T cell progenitors within the thymus, or their proliferation and differentiation require integrin-mediated adhesion.

Kindlins are a family of β integrin tail binding proteins that regulate the affinity of integrins for their ligands and induce signal transduction pathways downstream of activated, ligand-bound integrins (*Moser et al., 2009b*). Kindlins consist of 3 members: kindlin-1,–2, and −3; while kindlin-1 and −2 are widely expressed, hematopoietic cells exclusively express kindlin-3 (*Ussar et al., 2006*; *Weinstein et al., 2003*). Consequently, loss of kindlin-3 impairs the functions of all integrins on hematopoietic cells studied so far and leads to severe defects affecting platelets, leukocytes, osteoclasts and hematopoietic stem and progenitor cells (HSPCs) (*Krüger et al., 2008*; *Moser et al., 2009a*; *Moser et al., 2008*; *Ruppert et al., 2015*; *Schmidt et al., 2011*). In line with genetic studies in mice, loss-of-function mutations in the human *KINDLIN-3/FERMT3* gene cause leukocyte adhesion deficiency type-III (LAD-III) syndrome, which is characterized by severe bleedings, infections and accumulation of HSPCs in the blood circulation (*Kuijpers et al., 2009*; *Malinin et al., 2009*; *Mory et al., 2008*; *Ruppert et al., 2015*; *Svensson et al., 2009*).

In the present study, we investigated T-lymphopoiesis in kindlin-3-deficient mice. We found that loss of kindlin-3 protein expression results in progressive thymus atrophy, which is mainly caused by impaired colonization of the vascularised thymus by BM-derived T cell progenitors during late embryogenesis and after birth. In contrast, however, colonization of the non-vascularized thymic primordium by kindlin-3-deficient FL-derived progenitors proceeded without kindlin-3, albeit less efficiently, due to the lower vascular shear flow in embryos. Within the thymus anlage, the proliferation rate of kindlin-3-deficient T cell populations was reduced, while differentiation into mature CD4 and CD8 T cells was unaffected. Thus, these findings clearly show the crucial role of integrins during T cell development. Specifically, in the absence of kindlin-3 only a weak integrin-mediated T cell adhesion can occur, which suffices resistance to low systemic shear forces and enables T cell progenitor homing early during development. However, at later time points during development, when vascular shear forces increase, kindlin-3 is critical to stabilize T cell adhesion on endothelial cells allowing T cell progenitor homing into the thymus.

## Results

### Loss of kindlin-3 protein leads to progressive thymus atrophy

Kindlin-3 is expressed in CD4/CD8 double negative (DN) and double positive (DP) T cells from wild-type (WT) thymi and SP CD4 and CD8 T cells from WT spleens (*Figure 1—figure supplement 1A*). To test whether kindlin-3 expression is required for thymopoiesis, we investigated thymus morphology and size in kindlin-3-deficient (*Fermt3*−/−) mice, which die of severe bleedings and anemia within the first week after birth (*Moser et al., 2008*). The morphology of *Fermt3*−/− thymi was normal and the size steadily increased until postnatal day (P) 3 (*Figure 1A,B*). However, while WT thymi continued to grow postnatally, *Fermt3*−/− thymi shrank and became atrophic (*Figure 1B*). Consistently, the cellularity of WT thymi continuously increased during development (*Figure 1C*), while *Fermt3*−/− thymi showed an increase in cellularity only until P3, followed by a dramatic decrease at P6 and P8 (*Figure 1C*). Despite the progressive increase in cellularity until P3 the number of cells was significantly lower in *Fermt3*−/− thymi compared to controls at all time points analyzed (*Figure 1C*). Importantly, we did not observe any compensatory expression of kindlin-1 or −2 in kindlin-3-deficient thymi (*Figure 1—figure supplement 1B*).

Hematoxylin/eosin (H/E) staining of thymic section at different stages of development revealed a similar formation of medullary islets in P0 WT and *Fermt3*−/− thymi that enlarged and fused at P3 to produce a large central medulla surrounded by the cortex (*Figure 1D*). In line with the normal thymic cyto-architecture at P3, *Fermt3*−/− thymi displayed normal cortico-medullary junctions (CMJ) that separate CD4/CD8 DP T cells in the cortex from SP CD4 and CD8 T cells in the medulla, and normal epithelial meshes and vascular networks (*Figure 1—figure supplement 1C*). At P6 and P8, however,

**Figure 1.** Kindlin-3 deficiency causes thymus atrophy. (A and B) Images from *Fermt3*[+/+] (upper panel) and *Fermt3*[-/-] (lower panel) thymi isolated at the indicated embryonic and postnatal stages. Scale bars 1 mm. (C) Total thymocyte numbers at indicated time points (E14.5 to P8). N(E14.5)=28/28; N (E16.5)=12/12; N(E18.5)=30/21; N(P3)=6/4; N(P8)=3/2. (D) Haematoxylin and Eosin staining of paraffin-embedded thymus sections. Scale bar 1 mm. (E and F) Frequencies of CD4 and CD8 double negative (DN), double positive (DP) and single positive CD4 and CD8 thymocytes isolated from P3 and P6 thymi. N(P3)=16/18; N(P6)=8/4. Bars indicate means ± standard errors. **p<0.01; ***p<0.001. See also *Figure 1—figure supplement 1*.

DOI: https://doi.org/10.7554/eLife.35816.002

The following figure supplement is available for figure 1:

**Figure supplement 1.** Kindlin-3 deficiency causes thymus atrophy.
DOI: https://doi.org/10.7554/eLife.35816.003

the atrophic *Fermt3*[-/-] thymi showed a progressive enlargement of the medulla and a strong reduction of the cortex suggesting an imbalanced development towards more mature SP T cells (*Figure 1D*). This observation was confirmed by flow cytometry showing that DN T cells containing the most immature intrathymic T cells were significantly reduced at P3 and P6 in *Fermt3*[-/-] thymi compared to controls. Notably, while the relative distribution of DP and SP CD4 and CD8 T cells

was still comparable between WT and *Fermt3*[-/-] thymi at P3, the relative numbers of DP T cells decreased and SP CD4 and CD8 T cells increased in P6 thymi of *Fermt3*[-/-] mice (*Figure 1E,F*). However, despite the shift towards SP T cell populations in P6 *Fermt3*[-/-] thymi, their absolute numbers were still significantly reduced compared to controls (*Figure 1—figure supplement 1D,E*). The presence of normal relative SP T cell numbers in *Fermt3*[-/-] thymi until P3 suggests that intrathymic T cell differentiation proceeds independently of kindlin-3. This observation was supported by the normal levels of T cell receptor (TCR), CD3 and costimulatory molecules such as CD5 and CD24 on DP and SP T cells (*Figure 1—figure supplement 1F*).

## Thymocyte proliferation depends on kindlin-3 expression

To test whether the number of apoptotic thymocytes is altered within the P3 thymus we immunostained thymic sections for cleaved caspase-3 (*Figure 2A*) and analysed thymocytes by flow cytometry after annexin V and 7-AAD (7-Aminoactinomycin) staining (*Figure 2B*). We found that neither the number (*Figure 2C*) nor the frequency (*Figure 2D*) of apoptotic cells (annexin V[+]/7-AAD[-]) was altered in *Fermt3*[-/-] thymi. Furthermore, the different T cell subsets isolated from WT and *Fermt3*[-/-] thymi showed a comparable annexin V staining (*Figure 2E*).

Next, we determined thymocyte proliferation by in vivo bromodeoxyuridine (BrdU) labelling, which revealed a significantly reduced number of BrdU positive cells in P3 *Fermt3*[-/-] thymi both by immunohistochemistry (*Figure 2F,H*) and flow cytometry (*Figure 2G,I*). Notably, reduced proliferation was detected in both the DP and the SP T cell populations (*Figure 2J*). Since thymocyte proliferation relies on proper contact with antigen presenting cells (APC), we tested kindlin-3-deficient T cells for their ability to form immune synapses (IS) with WT DCs. To this end we loaded mature DCs with MOG35-55 peptide and added splenic CD4 T cells from *Fermt3*[+/fl] and *Fermt3*[fl/fl] mice, which express a TCR transgene specific for the myelin oligodendrocyte glycoprotein (MOG) 35–55 peptide (*2D2*), a double-fluorescent reporter transgene for measuring Cre activity (mTmG) and the CD4-Cre transgene (*Bettelli et al., 2003*; *Moretti et al., 2013*; *Muzumdar et al., 2007*). After 30 min the T-DC contacts were analysed by immunofluorescence staining. Although *Fermt3*[-/-] T cells formed fewer contacts with DCs compared to control T cells, recruitment of LFA-1 and actin to the contact area as well as phospho-Tyrosine signals were indistinguishable between them suggesting that *Fermt3*[-/-] T cells form IS with DCs (*Figure 2—figure supplement 1*). To study APC-induced T cell proliferation in vitro, we co-cultured MOG35-55 peptide-loaded dendritic cells with CSFE-labelled control and kindlin-3-deficient T cells derived from *Fermt3*[fl/fl]/*CD4-Cre/2D2* mice, and measured CSFE dilution by flow cytometry. In line with the observation that *Fermt3*[-/-] T cells can form contacts with DCs, a comparable proliferative response was observed when these T cells were stimulated by DCs loaded with high concentration of MOG35-55 peptide or when a strong TCR signal was induced directly with anti-CD3/CD28 antibodies. In contrast, *Fermt3*[-/-] T cells failed to proliferate when dendritic cells were loaded with lower concentrations of MOG35-55 peptide (*Figure 2K*) indicating that defective integrin-mediated contact between kindlin-3-deficient T cells and APCs causes a reduced proliferative response contributing to thymus atrophy of kindlin-3-deficient mice.

## T cell progenitor homing to the postnatal thymus

The reduced CD4 and CD8 DN T cell numbers in *Fermt3*[-/-] thymi (*Figure 1E,F* and *Figure 1—figure supplement 1D,E*) point to an impaired T cell progenitor homing to the thymus. Upon homing of T cell progenitors to the thymus, they progress through four DN stages before developing into DP cells. To test whether loss of kindlin-3 affects the early and/or late stages of DN development, we separated lineage-positive thymic cells from lineage-negative (Lin[neg]) DN cells by flow cytometry and gated the latter cell population for high and low CD44 and c-kit expression into DN stage 1 and 2 (DN$_{1-2}$) and DN stage 3 and 4 (DN$_{3-4}$) cells, respectively. In parallel, we gated DN cells for high and low CD44 and CD25 expression to discriminate between the four DN stages. These experiments revealed that the Lin[neg] DN population was significantly reduced in P3 and P6 thymi and the DN1 and DN2 stage thymocytes were virtually absent from P3 *Fermt3*[-/-] thymi (*Figure 3A* and *Figure 3—figure supplement 1*). DN1 stage cells are the immediate intrathymic descendants of circulating hematopoietic progenitor cells (Lin[neg], c-kit[+], Sca-1[-] and Sca-1[+]), which accumulate in the blood circulation of *Fermt3*[-/-] mice suggesting that extravasation of *Fermt3*[-/-] T cell progenitors is impaired (*Figure 3B,C*). To test this hypothesis experimentally, we transferred FL cells from E14.5 WT and



**Figure 2.** Reduced thymocyte proliferation contributes to thymus atrophy of kindlin-3-deficient mice. (A) Staining for cleaved caspase-3 counterstained with Mayer´s haematoxylin on paraffin sections of *Fermt3*[+/+] and *Fermt3*[-/-] P3 thymi. Scale bar 100 µm. (B) Single-cell suspensions of thymocytes from control and *Fermt3*[-/-] P3 thymi were stained with the apoptotic (Annexin V) and dead cell (7-AAD) markers. Numbers within the FACS blots represent percent of cells within each quadrant. (C) Number of apoptotic cells per field of view (fov) observed in (A). N = 10. Percentage of early apoptotic cells

*Figure 2 continued on next page*

*Figure 2 continued*

(Annexin V$^+$, 7-AAD$^-$) in control and *Fermt3$^{-/-}$* thymi (**D**) and in distinct T cell subpopulations (CD4/CD8 DN, CD4/CD8 DP, CD4 single positive and CD8 single positive) (**E**). N = 7/10. (**F**) Sections of *Fermt3$^{+/+}$* and *Fermt3$^{-/-}$* P3 thymi stained for BrdU incorporation and counterstained with Mayer´s haematoxylin. Scale bar 100 μm. (**G**) BrdU incorporation in thymocytes analysed by flow cytometry. (**H**) Numbers of BrdU positive cells per field of view (fov) observed in (**F**). N = 8/6. (**I**) Percentage of proliferating, BrdU positive thymocytes measured by flow cytometry. N = 14. (**J**) Percentage of BrdU positive cells in distinct T cell subpopulations. N = 14. (**K**) CD4 T cells isolated from spleens of control *Fermt3$^{fl/fl}$/2D2* and *Fermt3$^{fl/fl}$/2D2/CD4Cre* mice were stained with CFSE and stimulated either with DCs loaded with different concentrations of MOG$_{35-55}$ peptide or primed with anti-CD3e/CD28 antibodies and PMA. Representative histograms show CSFE dilution. Red-lined histograms represent cells incubated with not-loaded DCs or no antibodies. Bars indicate means ± standard errors. \*\*p<0.01; \*\*\*p<0.001. See also *Figure 2—figure supplement 1*.
DOI: https://doi.org/10.7554/eLife.35816.004

The following figure supplement is available for figure 2:

**Figure supplement 1.** Reduced thymocyte proliferation contributes to thymus atrophy of kindlin-3-deficient mice.
DOI: https://doi.org/10.7554/eLife.35816.005

*Fermt3$^{-/-}$* embryos into RAG2-deficient (*Rag2$^{-/-}$*) mice, in which T cell development is blocked at the DN3 stage due to loss of V(D)J rearrangement (*Shinkai et al., 1992*) and analysed size, weight and cellularity of the chimeric thymus and T cell frequency in the peripheral blood 8 to 10 weeks later. The transfer of *Fermt3$^{-/-}$* FL cells failed to reconstitute T cell development in the thymus of *Rag2$^{-/-}$* recipient mice (*Figure 4A–C*), which also contained almost no DN$_{1-2}$ cells (*Figure 4D,E*). Interestingly, a very small number of *Fermt3$^{-/-}$* T cell progenitors entered the *Rag2$^{-/-}$* recipient thymus, differentiated into DN4 cells (*Figure 4D,F*) and produced a small number of circulating CD4 and CD8 T cells, which homed to the spleen of *Rag2$^{-/-}$* mice (*Figure 4G–K*) indicating that kindlin-3 is very important for T cell progenitor homing to the adult thymus, but once in the thymus the progenitor cells differentiate into mature SP T cells that are released into the circulation.

Since kindlin-3-deficient HSCs show an approximately 5-fold reduced homing capacity to the bone marrow, which might eventually impact on T cell progenitor generation (*Ruppert et al., 2015*), we conditionally deleted *Fermt3* by injecting polyIC into *Fermt3$^{fl/fl}$/Mx1-Cre* mice and detected almost no DN (Lin$^{neg}$) cells in their thymi, whereas control thymi from polyIC-treated *Fermt3$^{fl/fl}$* mice clearly contained a discrete population of DN$_{1-2}$ stage T cells (*Figure 4—figure supplement 1A*). In addition, we generated mixed FL cell chimeras by transferring nine times more *Fermt3$^{-/-}$* FL cells than WT FL cells into lethally irradiated recipient mice and compared their homing capacity to the thymus. To discriminate between both cell populations the WT FL cells were isolated from B6.SJL embryos, which express the CD45.1 allelic variant, while the *Fermt3$^{-/-}$* embryos are from a C57BL/6 background expressing CD45.2. Despite the excess in transferred kindlin-3-deficient FL cells, only around 25% of the DN cells were of CD45.2 origin. As a control, chimeras made from 90% WT CD45.2 cells and 10% WT B6.SJL CD45.1 cells showed the 9:1 ratio in DN and all later T cell stages (*Figure 4—figure supplement 1B*). Moreover, to exclude a defect in a non-hematopoietic cell type as cause for the *Fermt3$^{-/-}$* T cell progenitor homing defect, we analysed thymi from Kindlin-3-EGFP knockin mice, which express kindlin-3 tagged C-terminally with EGFP. Flow cytometry of their thymi revealed that only CD45$^+$ leukocytes were EGFP-positive and that CD31$^+$ endothelial cells and EPCAM$^+$ thymic epithelial cells were EGFP negative (*Figure 4—figure supplement 2*). Altogether, these experiments show that kindlin-3-deficient T cell progenitors have a strong defect in homing to the postnatal and adult thymus and that the homing defect is of hematopoietic origin.

## T cell progenitor homing to the thymus primordium

Since thymi of newborn *Fermt3$^{-/-}$* mice lacked DN1 and DN2 cells but contained DN3 and DN4 cells (*Figure 3A*) we surmised that during development significant numbers of T cell progenitors from the FL must be able to colonize the fetal thymus. To test this hypothesis, we determined numbers and locations of CD45$^+$ hematopoietic progenitor cells in sections of the thymus primordium at different time points. In line with previous reports (*Douagi et al., 2000*; *Itoi et al., 2001*), we observed the first wave of WT CD45$^+$ progenitor cells within the perithymic mesenchyme at E11.5, and initial migration into the thymus primordium at E12.0 and scattering throughout the thymus at E12.5 and E13.5 (*Figure 5A–F*). Interestingly, also *Fermt3$^{-/-}$* progenitor cells could be detected in the thymus primordium and the surrounding mesenchyme, however at lower numbers (*Figure 5A–E*). Despite the reduction, CD45$^+$ progenitor cells accumulated at E13.5 in the *Fermt3$^{-/-}$* thymus (*Figure 5D*) and continued



**Figure 3.** Reduced frequencies of T cell progenitors in *Fermt3*[-/-] thymi. (**A**) Thymocytes from *Fermt3*[+/+] and *Fermt3*[-/-] mice were stained for lineage markers (B220, CD19, TER119, NK1.1, CD11b, Gr-1, CD8α, CD3e, TCRβ, TCRγδ and CD11c), CD44 and c-kit to identify $DN_{1-2}$ (Lin[neg], c-kit[hi], CD44[hi]) and the $DN_{3-4}$ (Lin[neg], c-kit[low], CD44[low]) populations. (**B and C**) PB from *Fermt3*[+/+] and *Fermt3*[-/-] animals (P3) were stained for lineage markers (TER119, B220, CD11b, Gr-1, CD11c, NK1.1, CD4 and CD8α), c-kit and Sca-1 to identify hematopoietic progenitor cells and analysed by flow cytometry. N = 4. Numbers within the representative FACS plots indicate cell percentages. Bars indicate means ± standard errors. **p<0.01; ***p<0.001. See also *Figure 3—figure supplement 1*.

DOI: https://doi.org/10.7554/eLife.35816.006

The following figure supplement is available for figure 3:

**Figure supplement 1.** Reduced frequencies of T cell prognitors in *Fermt3*[-/-] thymi.

DOI: https://doi.org/10.7554/eLife.35816.007

developing into $DN_{1-2}$ cells (*Figure 5F*). Strikingly, a dramatic drop in the $DN_{1-2}$ population was observed in *Fermt3*[-/-] thymi at E15-16 (*Figure 5G,H* and *Figure 5—figure supplement 1A,B*).

Next we tested whether an abnormal *Fermt3*[-/-] progenitor cell development in the FL or release from the FL into the circulation caused the reduced colonization of the thymus primordium. The flow cytometric analyses revealed that the absolute numbers of Lin[neg], c-kit[hi], CD45[+], PIR[+] T cell progenitors were comparable between WT and *Fermt3*[-/-] FLs, while their frequency, due to a decreased cellularity of *Fermt3*[-/-] FLs, was increased (*Figure 6A–C*). The numbers of circulating WT and *Fermt3*[-/-] Lin[neg], c-kit[hi], PIR[+] T cell progenitors increased between E11.5 to E12. However, while the frequency



**Figure 4.** Kindlin-3-deficient T cell progenitors are severely impaired in seeding the adult thymus. (A–C) Images (A), weights (B) and cell numbers (C) of thymi from adult WT and *Rag2*[-/-] mice and *Rag2*[-/-] mice 10 weeks after reconstitution with *Fermt3*[+/+] and *Fermt3*[-/-] fetal liver cells. Scale bar 5 mm. N = 10. (D) Single-cell suspensions were stained for lineage markers (B220, CD19, TER119, NK1.1, CD11b, Gr-1, CD11c, CD8α, CD3e, TCRβ, and TCRγδ), c-kit, CD44 and CD25 to identify the $DN_{1-2}$ ($Lin^{neg}$, $c-kit^{hi}$, $CD44^{hi}$) and the $DN_{3-4}$ ($Lin^{neg}$, $c-kit^{low}$, $CD44^{low}$) populations (upper panel), and DN1 ($Lin^{neg}$, $CD44^{hi}$, $CD25^-$), DN2 ($Lin^{neg}$, $CD44^{hi}$, $CD25^+$), DN3 ($Lin^{neg}$, $CD44^{low}$, $CD25^+$) and DN4 ($Lin^{neg}$, $CD44^{low}$, $CD25^-$) populations (lower panel) by flow cytometry. Frequencies of $DN_{1-2}$ (E) and DN4 (F) thymocytes. N = 8–10. (G) Analysis of peripheral blood T-lymphocytes. (H and I) Frequencies of circulating $CD4^+$ and $CD8^+$ T cells calculated from flow cytometric analyses shown in (G). N = 10. Numbers within the representative FACS plots indicate cell percentages. (J) Expression of talin-1 and kindlin-3 in $CD4^+$ T cells isolated from the spleen of *Rag2*[-/-] mice after reconstitution with *Fermt3*[+/+] and *Fermt3*[-/-] fetal liver cells. GAPDH serves as loading control. (K) genomic PCR of $CD8^+$ T cells sorted from the spleen of *Rag2*[-/-]/*Fermt3*[+/+] and *Rag2*[-/-]/*Fermt3*[-/-] chimeras. Numbers within the representative FACS plots indicate cell percentages. Bars indicate means ± standard errors. **p<0.01; ***p<0.001. See also *Figure 4—figure supplements 1* and *2*.

DOI: https://doi.org/10.7554/eLife.35816.008

The following figure supplements are available for figure 4:

**Figure supplement 1.** Kindlin-3-deficient T cell progenitors are severely impaired in seeding the adult thymus.
DOI: https://doi.org/10.7554/eLife.35816.009
**Figure supplement 2.** Kindlin-3-deficient T cell progenitors are severely impaired in seeding the adult thymus.
DOI: https://doi.org/10.7554/eLife.35816.010

of WT T cell progenitors steadily decreased between E12 and E13.5, the T cell progenitors accumulated in the circulation of *Fermt3*[-/-] fetuses due to their reduced extravasation from pharyngeal vessels and/or impaired migration towards the thymus primordium (*Figure 6D,E*).

T cell progenitor homing to the thymus is directed by chemokine gradients produced by thymic epithelial cells (*Koch and Radtke, 2011*). Therefore, we decided to test whether *Fermt3*[-/-] progenitors are able to sense the chemokine gradient released by thymic lobes in vitro and to invade the thymus. To this end, thymic lobes from WT E15.5 SJL embryos (CD45.1[+]) were depleted from their

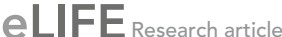

**Figure 5.** Kindlin-3-independent colonization of the fetal thymus. (A–D) Sagittal sections of whole embryos at gestational ages E11.5 (A), E12 (B), E12.5 (C) and E13.5 (D) were stained for fibronectin (perithymic mesenchyme, blue), cytokeratin (thymic primordium, green) and CD45 (potential T cell progenitor cells, red). Scale bars 100 μm. (E) Numbers of CD45$^+$ cells within the thymic primordium and in the perithymic mesenchyme at the indicated time points. N = 3–4. (F) Representative flow cytometric profiles of fetal thymocytes from control and *Fermt3$^{-/-}$* thymi at the indicated time points stained for lineage markers (B220, CD19, TER119, NK1.1, CD11b, Gr-1, CD11c, CD8α, CD3e, TCRβ, and TCRγδ), c-kit and CD44 to identify the DN$_{1-2}$ (Lin$^{neg}$, c-kit$^{hi}$, CD44$^{hi}$) and the DN$_{3-4}$ (Lin$^{neg}$, c-kit$^{low}$, CD44$^{low}$) populations. Numbers within the representative FACS plots indicate cell percentages. Frequencies (G) and total number (H) of DN$_{1-2}$ cells measured in the flow cytometric analyses shown in (F). N = 7–10 Bars indicate means ± standard errors. *p<0.05; **p<0.01; ***p<0.001. See also *Figure 5—figure supplement 1*.

DOI: https://doi.org/10.7554/eLife.35816.011

The following figure supplement is available for figure 5:

**Figure supplement 1.** Kindlin-3-independent colonization of the fetal thymus.

DOI: https://doi.org/10.7554/eLife.35816.012

T cells with deoxyguanosine (dGuo) and then positioned in a collagen matrix in close proximity to FL-derived Lin$^{neg}$ cells from WT or *Fermt3$^{-/-}$* E14.5 C57BL/6 (CD45.2$^+$) embryos (*Figure 7A*). Progenitor migration towards the thymic lobes was monitored for 36 to 48 hr by time-lapse video microscopy, which revealed that WT as well as *Fermt3$^{-/-}$* FL cells moved persistently towards the thymic lobes, while migration was not induced when thymic lobes were absent (*Figure 7B*). To further test whether FL-derived T cell progenitor cells had invaded the thymus capsule and developed into mature T cells ex vivo, the lobes were recovered from the collagen matrix and cultured for additional 18 days under conventional fetal thymus organ culture (FTOC) conditions (*Figure 7C*). Consistent



**Figure 6.** Fetal liver-derived T cell progenitors accumulate in the blood. (**A**) Fetal liver cells from control and *Fermt3*[-/-] embryos from indicated gestational stages were stained for lineage markers (B220, CD19, TER119, NK1.1, Gr-1 and Thy1.2), c-kit, CD45 and PIR and analysed by flow cytometry. Frequencies (**B**) and total numbers (**C**) of PIR[+] T cell progenitors per fetal liver. N(E12.5)=7/6; N(E13.5)=8/5; N(E14.5)=10/11; N(E15.5)=4/7. (**D**) FB from *Fermt3*[+/+] and *Fermt3*[-/-] embryos were stained for lineage markers (B220, CD19, TER119, NK1.1, Gr-1 and Thy1.2), c-kit and PIR and analysed by flow
*Figure 6 continued on next page*

*Figure 6 continued*

cytometry. (**E**) Frequencies of circulating PIR⁺ T cell progenitors determined in (**D**). N(E11.5)=5; N(E12)=8/3; N(E12.5)=4/5; N(E13.5)=7/6. Numbers within the representative FACS plots indicate cell percentages. Bars indicate means ± standard errors. *p<0.05; **p<0.01; ***p<0.001.

DOI: https://doi.org/10.7554/eLife.35816.013

with our in vivo data, CD45.2⁺ *Fermt3⁻/⁻* donor cells normally developed into DP, CD4 and CD8 SP cells (*Figure 7D*).

 Altogether these findings indicate that *Fermt3⁻/⁻* T cell progenitors normally develop in the FL, are released into the circulation, sense the chemokine gradient, leave the pharyngeal vessel, albeit

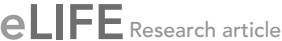

**Figure 7.** Kindlin-3-deficient T cell progenitors home to the fetal thymus and further develop into SP T cells ex vivo. (**A**) Schematic description of the thymus attraction experiment. (**B**) Representative images of the thymus attraction experiment at the beginning and 40 hr after starting the experiment. Lin^neg fetal liver cells from control or *Fermt3⁻/⁻* embryos (both CD45.2⁺) are seen in the upper right corner of the images, a dGuo-treated wild-type fetal thymus lobe (FT, CD45.1⁺) is located at the lower left corner. Scale bar 250 μm. (**C**) Schematic description of the fetal thymus organ culture. (**D**) dGuo-treated fetal thymus lobes seeded with fetal liver cells as performed in (**B**) were further cultured for 18 days. Cells were stained for CD45.1, CD45.2, CD4, and CD8. Numbers within the representative FACS plots indicate cell percentages.

DOI: https://doi.org/10.7554/eLife.35816.014

less efficiently than WT T cell progenitors, migrate towards the thymus primordium, invade the thymus capsule and differentiate into different T cell subsets.

## Blood flow velocity modulates *Fermt3*[-/-] T cell progenitor adhesion to vascular walls

The differential efficacy of *Fermt3*[-/-] T cell progenitor homing to the avascular and vascularized thymus was neither due to different kindlin-3 and talin-1 protein levels in lysates from sorted FL-derived T progenitor cells (Lin[neg], c-kit[+], PIR[+]) isolated from E13.5 embryos and DN1 cells (Lin[neg], CD44[+], CD25[-]) isolated from P6 thymi nor from an upregulation of talin-1 in *Fermt3*[-/-] PIR[+] T cell progenitors (*Figure 8A,B*). Flow cytometry also excluded differences in surface expression of α4, αL, β1, β2, β3 and β7 integrins on FL-derived T progenitor cells (Lin[neg], c-kit[+], PIR[+]) (*Figure 8—figure supplement 1*) and BM-derived LSK cells (*Ruppert et al., 2015*). Similarly, the CD31-positive vascular network was indistinguishable in E13.5 and E15.5 thymi from WT and *Fermt3*[-/-] mice excluding the possibility that a potential kindlin-3 expression in endothelial cells (*Bialkowska et al., 2010*) accounts for the defective homing to the vascularized thymus (data not shown). Since adhesion and extravasation of *Fermt3*[-/-] T cells can occur on inflamed endothelium expressing high levels of VCAM-1 and ICAM-1 (*Manevich-Mendelson et al., 2009*; *Moretti et al., 2013*), we determined the expression of VCAM-1, ICAM-1 and ICAM-2 by immunostaining and flow cytometry and found them even lower expressed in pharyngeal vessels compared to intrathymic vessels. Thus this also excludes that an elevated adhesion molecule expression on pharyngeal vessels compared to intrathymic vessels accounts for the better adhesion and extravasation (*Figure 8—figure supplement 2A–E*).

Integrin-mediated adhesion of T cells to endothelial adhesion molecules is influenced by the existing vascular shear stress (*Moretti et al., 2013*). We therefore hypothesized that *Fermt3*[-/-] T cell progenitors show impaired adhesion to the vascular endothelium when blood flow velocity and shear forces increase during intrauterine development. To test this hypothesis, we injected red fluorescent microbeads into the yolk-sac vasculature of E12.5 to E16.5 embryos and measured blood flow velocity by intravital microscopy. We found an increase in blood flow velocity (*Figure 8C*) and shear rates (*Figure 8D*) between E12.5 to 16.5. Accordingly, in vitro flow chamber assays revealed that isolated T cell progenitors (Lin[neg], c-kit[+], CD45[+], Pir[+]) hardly adhered to P-selectin, CCL21/CCL25 and ICAM-1 (*Figure 8E*), while they readily adhered to P-selectin, CCL21/CCL25 and VCAM-1 in a shear stress-dependent manner. Interestingly, *Fermt3*[-/-] T cell progenitors adhered to VCAM-1 but displayed a significantly reduced shear force resistance (*Figure 8F*) suggesting that the increase in blood flow velocity and shear forces during fetal development prevents *Fermt3*[-/-] T cell progenitor adhesion and extravasation at intrathymic vessels. To demonstrate that a strong reduction in kindlin-3 expression only allows adhesion of T cells to the vascular surface at sites with lower shear rates, we adoptively transferred a 1:1 ratio of CSFE and Far Red-labelled CD4 T cells from spleens and LNs of WT and *Fermt3* hypomorphic mice (K3n/-), respectively, into recipient mice and analysed their adhesion to the popliteal LN vasculature by spinning disc confocal microscopy (*Figure 8G,H*). *Fermt3* hypomorphic mice express only 5% kindlin-3 protein and therefore show a strong defect in leukocyte adhesion (*Klapproth et al., 2015*). As expected, we observed a reduced number of adherent *Fermt3* hypomorphic T cells in the LN vasculature compared to WT cells (*Figure 8H,I*). We then injected fluorescent microspheres and measured the blood flow velocities in LN vessel segments and determined shear rates adherent cells were exposed to in those vessels. We found that *Fermt3* hypomorphic cells adhered preferentially in vessel segments where blood flow velocity and shear rates were lower compared to WT T cells. The latter adhered to vessel segments with higher blood flow velocities and shear rates (*Figure 8J,K*). These findings indicate that kindlin-3 is crucial to stabilize integrin-mediated T cell adhesion to vessel walls exposed to high vascular shear forces.

## Discussion

In the present study, we used mice lacking the essential integrin regulatory protein kindlin-3 to address the role of integrin-mediated adhesion and signalling during T cell progenitor homing to the thymus and subsequent differentiation within the thymus. We found that in the absence of kindlin-3 a well-organized thymus with normal thymus architecture consisting of the four major compartments (subcapsular zone, cortex, medulla and cortico-medullary junction) is initially formed. The presence and clear separation of DP thymocytes in the cortex from SP CD4 and CD8 T cells in the



**Figure 8.** Kindlin-3 is important to stabilize the adhesion of T cell progenitors and mature T cells to vascular integrin ligands when blood flow velocities and shear rate levels increase during development and in vessel segments of higher blood flow within lymph nodes. (**A**) Talin and kindlin-3 expression in FACS-sorted Pir+ T cell progenitor cells from WT FL of E13.5 embryos compared to DN1 cells sorted from the thymus of P6 mice. GAPDH served as loading control. (**B**) Talin and kindlin-3 expression in control and kindlin-3$^{-/-}$ Pir+ T cell progenitor cells isolated from E13.5 FL. GAPDH served as loading control. Mean blood flow velocities (**C**) and shear rates (**D**) within the yolk sac vasculature of E12.5 to E16.5 embryos were determined by intravital microscopy. N = 14/3/8/21/18. (**E and F**) Relative adhesion of Pir+ T cell progenitor cells FACS-sorted from the FLs of control and *Fermt3*$^{-/-}$ E13.5 embryos on ICAM-1, P-selectin, CCL21 and CCL25 or on VCAM-1, P-selectin, CCL21 and CCL25 coated ibidi flow chambers with stepwise increasing shear rates. N ≥ 5. Bars indicate means ± standard deviation. (**G**) Purity of CD4+ T cells from WT and *Fermt3* hypomorphic (n/-) mice that have been labelled with CFSE and Far Red and mixed in a 1:1 ratio. Grey line represents isotype control. (**H,I**) Adhesion of CD4+ T cells in vivo. (**H**) Representative microscopic images of adherent (+/+, red) and (n/-, green) cells in the lymph node vasculature after adoptive transfer. Sum intensity Z projections of confocal stacks are shown. Segmented lines indicate vessel outlines. Scale bar = 50 μm. (**I**) Quantification of adherent CD4+ T cells (N = 18–19 vessels from three mice). (**J, K**) Microvascular blood flow in the lymph node vasculature. (**J**) Centerline blood flow velocity and (**K**) vascular shear rate in LN

*Figure 8 continued on next page*

*Figure 8 continued*

microvessel segments (N = 25–27 field of views from three mice). Bars indicate means ± standard deviation. **p<0.01; ***p<0.001. See also *Figure 8—figure supplements 1* and *2*.

DOI: https://doi.org/10.7554/eLife.35816.015

The following figure supplements are available for figure 8:

**Figure supplement 1.** Kindlin-3 is important to stabilize the adhesion of T cell progenitors and mature T cells to vascular integrin ligands when blood flow velocities and shear rate levels increase during development and in vessel segments of higher blood flow within lymph nodes.

DOI: https://doi.org/10.7554/eLife.35816.016

**Figure supplement 2.** Kindlin-3 is important to stabilize the adhesion of T cell progenitors and mature T cells to vascular integrin ligands when blood flow velocities and shear rate levels increase during development and in vessel segments of higher blood flow within lymph nodes.

DOI: https://doi.org/10.7554/eLife.35816.017

medulla of postnatal thymi indicate that chemokine-driven migration and positioning of developing T cells to specialized thymic microdomains is not impaired in the absence of kindlin-3 allowing a well-orchestrated developmental maturation from T cell progenitors to mature T cells. Thus, our data indicate that chemokine sensing and leukocyte migration within a 3D environment can occur in the absence of kindlin-3´s role to regulate integrin activity, which is in line with a previous report (*Lämmermann et al., 2008*).

While early thymic development is hardly affected by the lack of kindlin-3, the thymi from kindlin-3 deficient mice began shrinking after birth. We found two main reasons for the thymic atrophy. First, kindlin-3-deficient thymocytes exhibited reduced proliferation rates due to a weakened interaction with thymic APCs. A weakened interaction with APCs was also reported for CD4 T cells isolated from LFA-1 and talin-1 knockout mice to cause reduced T cell proliferation (*Graf et al., 2007*; *Kandula and Abraham, 2004*; *Wernimont et al., 2011*).

Secondly, we observed a severe T cell progenitor homing defect, which becomes apparent at the end of embryogenesis. T cell development in the thymus requires a periodic homing of FL or BM-derived T cell progenitors. The trafficking routes taken by T cell progenitors to the thymus depend on the vascularisation state of the thymus. After vascularization at E15/E16 (*Liu et al., 2006*), FL and later on BM-derived lymphoid progenitors directly enter the thymus at large post-capillary venules at the thymus cortico-medullary junction. This process is critically dependent on kindlin-3. Furthermore, our findings also show that kindlin-3-dependent adhesiveness through multiple integrin classes expressed on T lymphoid progenitors is essential for their homing to the vascularized thymus, while loss of individual integrins either did not or only partially diminish precursor T cell homing (*Scimone et al., 2006*).

Our findings also revealed that kindlin-3-deficient T cell progenitors were able to colonize the non-vascularized thymus primordium and develop into SP CD4 and CD8 T cells that are released into the circulation and found in the spleen. Although early colonization of the thymus anlage was slightly delayed and less efficient, kindlin-3-deficient T cell progenitors were capable to leave the pharyngeal vessels and migrate through the mesenchymal tissue before they reach and enter the thymic anlage. The reduced homing performance of kindlin-3-deficient T cell progenitors was neither due to their reduced production in the FL and/or release into the circulation, nor due to a defect in sensing chemokine cues that direct them towards the thymic primordium. So, we assume that reduced adhesion to the pharyngeal vessel wall accounts for the mild T cell progenitor homing defect of kindlin-3-deficient embryos, an assumption, which is also supported by the accumulation of T cell progenitors in the embryonic circulation.

Why can T cell progenitors extravasate from pharyngeal vessels at early developmental stages but fail to do so from post-capillary venules at the cortico-medullary junction of the vascularized thymus? We cannot fully exclude that differences in the expression of other cell surface receptors, such as CD44 or selectin ligands, may exist between FL and BM-derived T cell progenitors, which allow kindlin-3/integrin independent adhesion and extravasation. However, since impaired adhesion and extravasation becomes evident already before E16, a time point at which BM-derived T cell progenitors have not formed yet, we do not believe that this can explain the homing defect. The fact that FL-derived T cell progenitors are initially capable but later fail to home to the thymus suggests that cell extrinsic rather than cell intrinsic factors are responsible for the impaired colonization. Differences in thymus vascularization might be an obvious possibility, especially since a previous report

suggested kindin-3 expression in endothelial cells (*Bialkowska et al., 2010*). However, we found normal vascularization of the thymus anlage in the absence of kindlin-3 and could show that thymic endothelial cells do not express kindlin-3. Since kindlin-3-deficient thymi exhibited a dramatic drop in the number of DN1 and DN2 cells between E15 to E16, which is the time point when the thymus becomes vascularized (*Liu et al., 2006*), we suggest that differences in the adhesive properties of pharyngeal vs. intrathymic vessels account for this phenomenon. In support of this hypothesis, we reported that kindlin-3-deficient effector T cells can leave the brain vascular system during inflammation when the endothelium expresses ICAM-1 and VCAM-1 at high levels (*Moretti et al., 2013*). Similarly, kindlin-3-deficient effector T cells cannot extravasate from inflamed dermal microvessels but are able to leave postcapillary venules of skin-draining inflamed lymph nodes, probably because the high expression of integrin ligands and P-selectin stabilizes the weak adhesions of kindlin-3-deficient T cells (*Cohen et al., 2013*). However, expression of the integrin ligands ICAM-1/2 and VCAM-1 in pharyngeal vessels is lower than in intrathymic vessels thereby excluding this possibility. Therefore, we reasoned that another mechanism must be responsible for the homing ability of kindlin-3-deficient T cell progenitors to the early thymus.

Beside its role in triggering integrin activation in association with talin-1 (*Ma et al., 2008*; *Montanez et al., 2008*; *Moser et al., 2009a*; *Moser et al., 2009b*; *Moser et al., 2008*), kindlin-3 is important in stabilizing integrin-ligand interactions by promoting integrin clustering (*Feng et al., 2012*; *Ye et al., 2013*). Thus, the gradual T cell progenitor homing defect observed in kindlin-3-deficient mice might be due to instable T cell progenitor/endothelial adhesions, which are unable to withstand increasing shear forces elicited by the maturing blood circulation. Using an intravital microscopy approach we indeed measured an increase in systemic blood flow velocities and shear rates leading to higher hemodynamic forces T-cell progenitors are exposed to during development. In addition, our flow chamber experiments with FL-derived T cell progenitors confirmed this as kindlin-3-deficient cells could adhere to VCAM-1 coated surfaces at low shear stress levels but readily detached at higher shear stress levels when compared to control cells. Moreover, T cells expressing very low kindin-3 levels predominantly adhere to LN vessel segments with lower shear rates than WT T cells. Thus, low shear force conditions may allow adhesion of kindlin-3-deficient T cell progenitors to fetal pharyngeal microvessels, hence enabling their extravasation. However with increasing shear forces at later time points during fetal life, T cell progenitors lacking kindlin-3 and therefore the ability to strengthening integrin-mediated adhesion are not able to efficiently adhere to and migrate out of microvessels into the thymus. The ability to resist low shear forces is also supported by an early report that found high expression of α4β1 on the surface of FL-derived T cell progenitors (*Kawakami et al., 1999*). In addition, we and others have shown that α4β1-mediated adhesiveness to VCAM-1 is less affected by loss of kindlin-3 in contrast to αLβ2 integrin-mediated adhesion to ICAM-1 (*Klapproth et al., 2015*; *Manevich-Mendelson et al., 2009*; *Moretti et al., 2013*). In this respect, it is interesting to note that lack of β1 integrin expression on hematopoietic stem cells (HSC) abolishes their extravasation and homing to the BM (*Potocnik et al., 2000*), whereas kindlin-3-deficient HSCs revealed significantly diminished but not absent homing to the BM (*Ruppert et al., 2015*).

Taken together, our work demonstrates that kindlin-3 is dispensable for integrin-mediated T cell adhesion and signalling at low and indispensable at high vascular shear forces. Furthermore, we also show a general, crucial role of integrins during T cell development and suggest an additional not yet investigated T cell phenotype in LADIII patients.

## Materials and methods

### Key resources table

| Reagent type (species) or resource | Designation | Source or reference | Identifiers | Additional information |
| --- | --- | --- | --- | --- |
| Genetic reagent (*M.musculus*) | *Fermt3*[-/-] | PMID: 18278053 | RRID: MGI:2147790 | Dr. Reinhard Fässler (Max Planck Institute of Biochemistry) |
| Genetic reagent (*M.musculus*) | *Fermt3*[flox] | PMID: 24089451 | RRID: MGI:5551978 | Dr. Reinhard Fässler (Max Planck Institute of Biochemistry) |

*Continued on next page*

*Continued*

| Reagent type (species) or resource | Designation | Source or reference | Identifiers | Additional information |
|---|---|---|---|---|
| Genetic reagent (*M.musculus*) | *Fermt3$^{n/-}$* | PMID: 26438512 | RRID: MGI:3785479 | Dr. Reinhard Fässler (Max Planck Institute of Biochemistry) |
| Genetic reagent (*M.musculus*) | *Rag2$^{-/-}$* | PMID: 1547487 | RRID: MGI:1858556 | Dr. Frederick W Alt (Howard Hughes Medical Institute) |
| Genetic reagent (*M.musculus*) | *Mx1-Cre* | PMID: 7660125 | RRID: MGI:2176073 | Dr. Klaus Rajewsky (Max Delbrück Center for Molecular Medicine) |
| Genetic reagent (*M.musculus*) | *CD4-Cre* | PMID: 11728338 | RRID: MGI: 2386448 | Dr. Christopher B. Wilson, (University of Washington) |
| Genetic reagent (*M.musculus*) | *2D2* | PMID: 12732654 | RRID: MGI:3700794 | Dr. Vijay K Kuchroo (Center for Neurologic Diseases) |
| Genetic reagent (*M.musculus*) | *mTmG* | PMID: 17868096 | RRID: MGI:3716464 | Dr. Liqun Luo (Howard Hughes Medical Institute) |
| Antibody | anti-B220-biotin | BD Pharmingen | Cat. #: 553085; RRID: AB_394615 | FACS (1:200) |
| Antibody | POD-coupled anti-BrdU | Roche | Cat. #: 11 585 860 001; RRID: AB_514485 | IHC (1:30) |
| Antibody | rabbit anti-mouse cleaved caspase-3 | Cell Signalling | Cat. #: 9661; RRID: AB_2341188 | IHC (1:100) |
| Antibody | anti-CD3e | eBioscience | Cat. #: 14-0031-82; RRID: AB_467049 | 10 µg/ml |
| Antibody | anti-CD3e-biotin | BD Pharmingen | Cat. #: 553060; RRID: AB_394593 | FACS (1:200) |
| Antibody | anti-CD3-PE | BD Pharmingen | Cat. #: 555275; RRID: AB_395699 | FACS (1:200) |
| Antibody | anti-CD4-biotin | BD Pharmingen | Cat. #: 553649; RRID: AB_394969 | FACS (1:200) |
| Antibody | anti-CD4-FITC | BD Pharmingen | Cat. #: 553651; RRID: AB_394971 | FACS (1:200) |
| Antibody | anti-CD4-PerCP | BD Pharmingen | Cat. #: 550954; RRID: AB_393977 | FACS (1:200) |
| Antibody | anti-CD5-PE | eBioscience | Cat. #: 12-0051-82; RRID: AB_465523 | FACS (1:200) |
| Antibody | anti-CD8α-biotin | BD Pharmingen | Cat. #: 553028; RRID: AB_394566 | FACS (1:200) |
| Antibody | anti-CD8-APC | eBioscience | Cat. #: 17-0081-81; RRID: AB_469334 | FACS (1:200) |
| Antibody | anti-CD8-PE | eBioscience | Cat. #: 12-0081-81; RRID: AB_465529 | FACS (1:200) |
| Antibody | anti-CD11b-biotin | BD Pharmingen | Cat. #: 557395; RRID: AB_2296385 | FACS (1:200) |
| Antibody | anti-CD11c-biotin | BD Pharmingen | Cat. #: 553800; RRID: AB_395059 | FACS (1:200) |
| Antibody | anti-CD16/CD32 | BD Pharmingen | Cat. #: 553142; RRID:AB_394657 | FACS (1:400) |
| Antibody | anti-CD19-biotin | eBioscience | Cat. #: 13-0193-82; RRID: AB_657656 | FACS (1:200) |
| Antibody | anti-CD24-PE | BD Pharmingen | Cat. #: 553262; RRID: AB_394741 | FACS (1:200) |

*Continued on next page*

*Continued*

| Reagent type (species) or resource | Designation | Source or reference | Identifiers | Additional information |
|---|---|---|---|---|
| Antibody | anti-CD25-FITC | BD Pharmingen | Cat. #: 553071; RRID: AB_394603 | FACS (1:200) |
| Antibody | anti-CD28 | eBioscience | Cat. #: 14-0281-82; RRID: AB_467190 | 2 µg/ml |
| Antibody | rat anti-CD31 (PECAM-1) | BD Pharmingen | Cat. #: 557355; RRID: AB_396660 | IHC (1:300) |
| Antibody | anti-CD31-Alexa Fluor 647 | Biolegend | Cat. #: 102415; RRID: AB_493411 | IHC (1:200) |
| Antibody | anti-CD31-eFLour450 | eBioscience | Cat. #: 48-0311-82; RRID: AB_10598807 | FACS (1:200) |
| Antibody | anti-CD44-PE | BD Pharmingen | Cat. #: 553134; RRID: AB_394649 | FACS (1:200) |
| Antibody | anti-CD45-FITC | eBioscience | Cat. #: 14-0451-82; RRID: AB_467251 | IHC (1:100) |
| Antibody | anti-CD45.2-biotin | BD Pharmingen | Cat. #: 553771; RRID: AB_395040 | FACS (1:200) |
| Antibody | anti-CD45.2-APC | eBioscience | Cat. #: 17-0454-82; RRID: AB_469400 | FACS (1:200) |
| Antibody | hamster anti-CD54 (ICAM) | BD Pharmingen | Cat. #: 553250; RRID: AB_394732 | IHC (1:100) |
| Antibody | goat anti-CD106 (VCAM) | R and D Systems | Cat. #: AF643; RRID: AB_355499 | IHC (1:80) |
| Antibody | anti-c-kit-APC | eBioscience | Cat. #: 17-1171-81; RRID: AB_469429 | FACS (1:200) |
| Antibody | anti-c-kit-PerCP | Biolegend | Cat. #: 105821; RRID: AB_893230 | FACS (1:200) |
| Antibody | mouse anti-pan-cytokeratin | Sigma-Aldrich | Cat. #: C2562; RRID: AB_476839 | IHC (1:400) |
| Antibody | anti-EpCAM-APC-eFlour780 | Thermo Fisher Scientific | Cat. #: 47-5791-82; RRID: AB_2573986 | FACS (1:200) |
| Antibody | rat anti-ER-TR4 | Eric Vroegindeweij (Princess Maxima center of pediatric oncology) | | IHC undiluted |
| Antibody | rat anti-ER-TR5 | Eric Vroegindeweij (Princess Maxima center of pediatric oncology) | | IHC undiluted |
| Antibody | anti-F4/80-biotin | Bio-Rad | Cat. #: MCA497GA; RRID: AB_323806 | FACS (1:200) |
| Antibody | alexa-fluor-488-conjugated anti-Fluorescin IgG | Thermo Fisher Scientific | Cat. #: A-11090; RRID: AB_221562 | IHC (1:500) |
| Antibody | rat anti-ICAM-1 PE | Biolegend | Cat. #:116107; RRID: AB_313698 | FACS (1:200) |
| Antibody | rabbit anti-Fibronectin | Merck Millipore | Cat. #: AB2033; RRID: AB_2105702 | IHC (1:300) |
| Antibody | mouse anti-GAPDH | Merck Millipore | Cat. #: CB1001; RRID: AB_2107426 | WB (1:20.000) |
| Antibody | anti-Gr-1-biotin | BD Pharmingen | Cat. #: 553125; RRID: AB_394641 | FACS (1:200) |
| Antibody | alexa-fluor-546-conjugated anti-goat IgG | Thermo Fisher Scientific | Cat. #: A-11056; RRID: AB_2534103 | IHC (1:300) |

*Continued on next page*

*Continued*

| Reagent type (species) or resource | Designation | Source or reference | Identifiers | Additional information |
|---|---|---|---|---|
| Antibody | alexa-fluor-546-conjugated anti-hamster IgG | Thermo Fisher Scientific | Cat: #: A-21111; RRID: AB_2535760 | IHC (1:300) |
| Antibody | alexa-fluor-546-conjugated anti-mouse IgG | Thermo Fisher Scientific | Cat: #: A-11003; RRID: AB_2534071 | IHC (1:300) |
| Antibody | alexa-fluor-647-conjugated anti-mouse IgG | Thermo Fisher Scientific | Cat: #: A-31571; RRID: AB_162542 | IHC (1:300) |
| Antibody | alexa-fluor-647-conjugated anti-rabbit IgG | Thermo Fisher Scientific | Cat: #: A-21244; RRID: AB_2535812 | IHC (1:300) |
| Antibody | alexa-fluor-488-conjugated anti-rat IgG | Thermo Fisher Scientific | Cat: #: A-11006; RRID: AB_2534074 | IHC (1:300) |
| Antibody | alexa-fluor-647-conjugated anti-rat IgG | Thermo Fisher Scientific | Cat: #: A-21247; RRID: AB_141778 | IHC (1:300) |
| Antibody | anti-Integrin α4-PE | BD Pharmingen | Cat: #: 553157; RRID: AB_394670 | FACS (1:200) |
| Antibody | anti-Integrin αL-PE | BD Pharmingen | Cat: #: 553121; RRID: AB_394637 | FACS (1:200) |
| Antibody | anti-Integrin β1-PE | Biolegend | Cat: #: 102207; RRID: AB_312884 | FACS (1:200) |
| Antibody | anti-Integrin β2-PE | BD Pharmingen | Cat: #: 553293; RRID: AB_394762 | FACS (1:200) |
| Antibody | anti-Integrin β3-PE | eBioscience | Cat: #: 12-0611-81; RRID: AB_465717 | FACS (1:200) |
| Antibody | anti-Integrin β7-PE | BD Pharmingen | Cat: #: 557498; RRID: AB_396735 | FACS (1:200) |
| Antibody | rabbit anti-Kindlin-1 | Markus Moser (MPI of Biochemistry) | | WB (1:1000) |
| Antibody | mouse anti-Kindlin-2 | Sigma-Aldrich | Cat. # SAB4200525-200UL | WB (1:1000) |
| Antibody | rabbit anti-Kindlin-3 | Markus Moser (MPI of Biochemistry) | | WB (1:3000) |
| Antibody | rabbit anti-pan-laminin | Rupert Timpl | | IHC (1:700) |
| Antibody | anti-LFA-1-PE | BD Pharmingen | Cat. #: 553121; RRID: AB_394637 | IF (1:100) |
| Antibody | anti-NK1.1-biotin | BD Pharmingen | Cat. #: 553163; RRID: AB_394675 | FACS (1:200) |
| Antibody | anti-PIR-APC | eBioscience | Cat. #: 17-3101-82; RRID: AB_1944406 | FACS (1:200) |
| Antibody | anti-PIR-PE | BD Pharmingen | Cat. #: 550349; RRID: AB_393628 | FACS (1:200) |
| Antibody | anti-p-Tyrosine | Santa Cruz Biotechnology | Cat. #: sc-7020; RRID: AB_628123 | IF (1:50) |
| Antibody | anti-Sca-1-PE | eBioscience | Cat. #: 12-5981-82; RRID: AB_466086 | FACS (1:200) |
| Antibody | mouse anti-talin | Sigma-Aldrich | Cat. #: T3287; RRID: AB_477572 | WB (1:20.000) |
| Antibody | anti-TCRβ-PE | eBioscience | Cat. #: 12-5961-82; RRID: AB_466066 | FACS (1:200) |

*Continued on next page*

*Continued*

| Reagent type (species) or resource | Designation | Source or reference | Identifiers | Additional information |
|---|---|---|---|---|
| Antibody | anti-TCRγδ-biotin | BD Pharmingen | Cat. #: 553176; RRID: AB_394687 | FACS (1:200) |
| Antibody | anti-Ter119-biotin | BD Pharmingen | Cat. #: 553672; RRID: AB_394985 | FACS (1:200) |
| Antibody | anti-Thy1.2-biotin | BD Pharmingen | Cat. #: 553001; RRID: AB_394540 | FACS (1:200) |
| Antibody | rat anti-VCAM-1 PE | Biolegend | Cat. #: 105713; RRID: AB_1134166 | FACS (1:200) |
| Peptide, recombinant protein | recombinant mouse CCL21 | R and D Systems | Cat. #: 457–6C; Accession #: P84444.1 | |
| Peptide, recombinant protein | recombinant mouse CCL25 | R and D Systems | Cat. #: 481-TK; Accession #: O35903.1 | |
| Peptide, recombinant protein | recombinant mouse ICAM-1-FC | R and D Systems | Cat. #: 796-IC; Accession #: Q3U8M7 | |
| Peptide, recombinant protein | MOG35-55 | MPI of Biochemistry, Core facility | | |
| Peptide, recombinant protein | recombinant mouse P-Selectin-FC | R and D Systems | Cat. #: 737-PS; Accession #: Q01102 | |
| Peptide, recombinant protein | recombinant mouse VCAM-1-FC | R and D Systems | Cat. #: 643-VM; Accession #: CAA47989 | |
| Commercial assay or kit | PE Annexin V Apoptosis Detection Kit | BD Biosciences | Cat. #: 559763 | |
| Commercial assay or kit | BrdU Flow Kit | BD Pharmingen | Cat. #: 559619; RRID: AB_2617060 | |
| Commercial assay or kit | anti-biotin MicroBeads | Miltenyi | Cat. #: 130-090-485; RRID: AB_244365 | |
| Commercial assay or kit | CD4 T cell isolation kit | Miltenyi | Cat. #: 130-104-454 | |
| Commercial assay or kit | Vectastain Elite ABC HRP Kit, rabbit IgG | VECTOR laboratories | Cat. #: PK-6101; RRID: AB_2336820 | |
| Chemical compound, drug | CellTrace CFSE Cell Proliferation Kit | Thermo Fisher Scientific | Cat. #: C34554 | |
| Chemical compound, drug | Collagenase D | Sigma-Aldrich | Cat. #: C0130 | 1 mg/ml |
| Chemical compound, drug | FITC-Dextran | Sigma-Aldrich | Cat. #: FD150S | |
| Chemical compound, drug | DNaseI | Sigma-Aldrich | Cat. #: D5025 | 100 U/ml |
| Chemical compound, drug | dGuo | Sigma-Aldrich | Cat. #: D0901 | 1.35 mM |
| Chemical compound, drug | CellTrace Far Red Cell Proliferation Kit | Thermo Fisher Scientific | Cat. #: C34564 | 1 µM |
| Chemical compound, drug | FluoSpheres 580/606 | Thermo Fisher Scientific | Cat. #: F8821 | 1 µM |
| Chemical compound, drug | LPS | Sigma-Aldrich | Cat. #: L2654 | 1 mg/ml |
| Chemical compound, drug | poly-L-Lysine | Sigma-Aldrich | Cat. #: P8920 | pure |

*Continued on next page*

*Continued*

| Reagent type (species) or resource | Designation | Source or reference | Identifiers | Additional information |
|---|---|---|---|---|
| Chemical compound, drug | Phalloidin Alexa350 | Thermo Fisher Scientific | Cat. #: A22281 | IF (1:40) |
| Chemical compound, drug | PureCol | Advanced BioMatrix | Cat. #: 5005–100 ML | |
| Chemical compound, drug | Streptavidin Cy3 | Jackson Immuno Research | Cat. #: 016-160-084; RRID: AB_2337244 | FACS (1:400) |
| Chemical compound, drug | Streptavidin FITC | Thermo Fisher Scientific | Cat. #: SA10002 | FACS (1:200) |
| Chemical compound, drug | Streptavidin PerCP-Cy5.5 | BD Pharmingen | Cat. #: 551419 | FACS (1:200) |
| Software, algorithm | ImageJ | National Institute of Health | RRID:SCR_003070 | |
| Software, algorithm | ImSpector Pro | LaVision BioTec GmbH | | |
| Software, algorithm | Photoshop | Adobe | RRID:SCR_014199 | |
| Software, algorithm | Slidebook 6.0.8 Software | Intelligent Imaging Innovations | RRID:SCR_014300 | |
| Software, algorithm | VirtualDub | GNU General Public License (GPL) | | |
| Other | Glass Bottom Microwell Dishes | MatTek Corporation | Cat. #: P35G-1.5–20 C | |
| Other | Ibidi flow chambers μ-Slide I 0.8 Luer IbiTreat | Ibidi | Cat. #: 80196 | |

## Mice and chimeras

*Fermt3*[-/-] mice have been described previously (*Moser et al., 2008*). All mice were from a mixed 129/SvJ x C57BL/6 background. FL cell chimeras were generated as previously described (*Prümmer and Fliedner, 1986*; *Tulunay et al., 1975*). Briefly, 5–6 × $10^6$ cells from the FL of E14.5 embryos were injected into the tail vain of sub-lethally irradiated (1 × 6 Gy) *Rag2*[-/-] mice (*Shinkai et al., 1992*) and chimeras were analyzed 8 to 10 weeks later. Mixed FL cell chimeras were generated by injection of 5 × $10^6$ E14.5 FL cells from WT or *Fermt3*[-/-] embryos, respectively, which were mixed with E14.5 FL cells from congenic B6.SJL WT embryos in a 9:1 ratio, into lethally irradiated C57BL/6 mice (1 × 7 Gy and 1 × 6 Gy with 4 hr in between). Conditional *Fermt3* deletion was obtained upon two intraperitoneal injections of pI-pC (500 μg/mouse) (GE Healthcare) into *Fermt3*[fl/fl]/*Mx1-Cre* mice. Pictures of dissected thymi were taken with a ProgRes C14 camera (JenOptic) connected to a Leica MZ FLIII stereomicroscope. All animals were mated and maintained under specific pathogen-free conditions at the animal facility of the Max Planck Institute of Biochemistry. All mouse experiments were performed with the approval of the District Government of Bavaria.

## Western blot analysis

Protein lysates were separated on a 10% SDS-PAGE and analyzed by Western blotting using polyclonal rabbit anti-mouse kindlin-1 and kindlin-3 antisera (*Ussar et al., 2008*; *Ussar et al., 2006*), monoclonal mouse anti-kindlin-2 (Sigma), monoclonal mouse anti-talin (Sigma) and mouse anti-GAPDH monoclonal antibody (Calbiochem) with a LAS-4000 imager (Fujifilm).

## Multi-color flow cytometry analysis and cell sorting

Multi-color flow-cytometric analysis and cell sorting were performed using FACS-Calibur and FACS-Aria flow cytometers (BD Biosciences), respectively. Data for viable cells, which were determined by forward light scatter and propidium iodide exclusion, were obtained with BD CellQuest Pro software. All antibodies we used were purchased from eBioscience or BD Pharmingen. Lineage negative (Lin[neg]) cell populations were obtained by removal of mature cells using the following cocktails of

biotinylated antibodies: anti-B220, anti-CD19, anti-TER119, anti-NK1.1, anti-CD11b (Mac-1), anti-Gr-1, anti-CD8α, anti-CD3e, anti-TCRβ, anti-TCRγδ and anti-CD11c for fetal and adult thymi; anti-TER119, anti-B220, anti-CD11b, anti-Gr-1, anti-CD11c, anti-NK1.1, anti-CD4 and-CD8α for postnatal peripheral blood; anti-B220, anti-CD19, anti-TER119, anti-NK1.1, anti-Gr-1 and anti-Thy1.2 for FL and fetal blood (FB). Cells were then stained either with PerCP-Cy5.5-conjugated or FITC-conjugated Streptavidin and excluded by FL3 or FL1 FACS channels. The resulting Lin^neg cell populations were further gated with anti-CD25-FITC, anti-CD44-PE and anti-c-kit-APC for thymocytes; with anti-Sca-1-PE and anti-c-kit-APC for postnatal peripheral blood cells; with anti-CD45-FITC, anti-PIR-PE and anti-c-kit-APC or anti-c-kit-PerCP, anti-PIR-APC and anti-integrin-PE (α4, αL, β1, β2, β3, β7) for FL cells; with anti-PIR-PE and anti-c-kit-APC for FB cells.

Mature T cells were analyzed for surface receptor expression with anti-CD4-FITC, anti CD8-APC, anti-CD3-PE, anti-CD5-PE, anti-CD24-PE and TCRβ-PE.

FL-Lin^neg cells used for the time-lapse visualization of thymus attraction (see below) were obtained by negative selection through MACS separation columns with anti-Biotin-coupled magnetic beads according to the manufacturer's instructions (MACS Miltenyi Biotec).

Expression of kindlin-3 in different thymic cell types was analysed by flow cytometry with thymi from Kindlin-3-EGFP knockin mice. A characterization of this mouse strain expressing a kindlin-3-EGFP fusion protein will be described elsewhere. In order to analyse kindlin-3 expression in leukocytes, thymic epithelial cells and endothelial cells, an adult thymus or a pool of 5 E15.5 fetal thymi were squeezed between two glass slides to remove the majority of T cells. The remaining tissue was digested in 1 mg/ml Collagenase D and 100 U/ml DNase I (both from Sigma) in 3% FCS/PBS for 20 to 30 min at 37°C, followed by a Fc-receptor block with anti-CD16/CD32 antibody and incubation with anti-CD31 eFlour450, anti-EpCAM APC eFlour780 (both Invitrogen), anti-CD45.2 APC and anti-CD3 PE antibodies.

## Immunofluorescence analysis of embryos and thymi

4% PFA-prefixed, frozen embryos and thymi embedded in Shandon Cryomatrix compound (Thermo Scientific) were first sliced into 8–10 µm thick sections and stained with the following antibodies for the embryos: rabbit anti-fibronectin polyclonal antibody (Chemicon), mouse anti-pan-cytokeratin monoclonal antibody (Sigma), rat anti-CD45-FITC (eBioscience), goat anti-CD106 (VCAM-1, R & D Systems), hamster anti-CD54 (ICAM-1, Pharmingen) and rat anti-CD31 (PECAM-1, Pharmingen) followed by Alexa-Fluor-647-conjugated anti-rabbit IgG, Alexa-Fluor-546-conjugated anti-mouse IgG, Alexa-Fluor-488-conjugated anti-rat IgG, Alexa-Fluor-647-conjugated anti-mouse IgG, Alexa-Fluor-546-conjugated anti-goat IgG, and Fluor-546-conjugated anti-hamster IgG antibodies (Invitrogen); for the thymus: rabbit anti-pan-laminin polyclonal antibody (gift from Rupert Timpl), mouse anti-pan-cytokeratin monoclonal antibody (Sigma), rat anti-CD31 (PECAM-1, Pharmingen), rat anti-CD4-FITC (eBioscience), rat anti-CD8-Biotin (eBioscience), anti-CD106 (VCAM-1, R & D Systems), hamster anti-CD54 (ICAM-1, Pharmingen), rat anti-ER-TR4 and anti-ER-TR5 polyclonal antibodies (kindly provided by Eric Vroegindeweij) followed by Alexa-Fluor-647-conjugated anti-rabbit IgG, Alexa-Fluor-647-conjugated anti-rat IgG, Alexa-Fluor-488-conjugated anti-Fluorescein/Oregon green IgG, Cy3-conjugated Streptavidin (Jackson), Alexa-Fluor-488-conjugated anti-rat IgG, Fluor-546-conjugated anti-goat IgG, and Fluor-546-conjugated anti-hamster IgG antibodies (Invitrogen). Thymic sections were fixed with acetone instead of PFA for anti-ER-TR4 and -TR5 staining.

Relative quantification of ICAM-1 and VCAM-1 expression was performed with directly labelled antibodies: rat anti-ICAM-1-PE, rat anti-VCAM-1-PE and the respective isotypes, rat anti-CD31-Alexa-Fluor 647 (all from Biolegend).

The stained sections were mounted with a fluorescence mounting medium (Elvanol) and images were acquired with a fluorescence microscope (Imager.Z1, Zeiss) and analyzed by the Axio Vision 40 (version 4.8.2.0) software (Zeiss), utilizing 10x, 20x and 40x magnifications.

## Immunohistologic analysis of the thymus

4% PFA fixed and paraffin-embedded thymi were sliced into 8 µm tick sections and stained with antibodies. The sections were counterstained with Mayer's haematoxylin (Merck) and mounted in Entellan (Merck). Haematoxylin and Eosin (H/E) staining was performed according to standard protocol.

Images were acquired with the light microscope Axioskop (Zeiss) and analyzed with the Adobe Photoshop software.

## Apoptosis assays

Paraffin-embedded thymus sections were stained with rabbit anti-mouse cleaved caspase-3 antibody (Cell Signalling). Apoptotic cells were detected by DAB staining after treatment with the immuno-peroxidase Vectastain ABC system (Vector Laboratories).

Thymocytes from P3 mice were stained with PE-labeled Annexin V and 7-AAD according to the manufacturer's instructions (BD Pharmingen Apoptosis Detection Kit). T cells were additionally stained for anti-CD4-FITC and anti-CD8-APC and analysed by FACS.

## Immune synapse analysis

CD4 T cells were purified from spleen by negative selection using biotinylated anti-CD8, anti-B220, anti-Gr-1, anti-F4/80 and anti-Ter119 antibodies and anti-biotin microbeads following the manufacturer's instructions (MACS Miltenyi Biotec). Bone marrow derived, LPS matured dendritic cells were loaded with 1 µg/ml MOG35-55 peptide for 2 hr at a concentration of $10^6$ cell/ml. $1 \times 10^5$ loaded DCs were mixed with $5 \times 10^5$ CD4 T cells in 200 µl R10 medium and seeded on poly-L-Lysine coated glass-bottom dishes (Mattek). Cells were fixed after 30 min by adding 200 µl warm 6% PFA in R10 medium for 20 min at 37°C. After permeabilization with 0.2% Triton X-100 in PBS for 15 min and blocking with 1% BSA for 1 h cells were stained with antibodies against LFA-1 (clone 2D7, BD Biosciences) and p-Tyrosine (PY199, Santa Cruz Biotechnology). Phalloidin Alexa350 was used to visualize actin. Cells were imaged with a Leica TCS SP5 X confocal microscope (Leica Microsystems) equipped with a $63 \times$ NA 1.40 oil objective and Leica Confocal Software (LAS AF). Single channels were imaged sequentially. All pictures were processed with Photoshop (Adobe Systems, San José, California, USA).

## Proliferation assays

For in vivo labelling of thymocytes, bromodeoxyuridine (BrdU; 50 µg/gram body weight) was intraperitoneally injected into P3 mice. One hour after injection, cells were isolated from thymi and stained with FITC-labeled anti-BrdU monoclonal antibody (BD Pharmingen BrdU Flow Kit), anti-CD4-PerCP and anti-CD8-PE antibodies, and analyzed by flow cytometry. Alternatively, paraffin-embedded thymus sections were DAB stained after incubation with a POD-coupled anti-BrdU antibody (Roche).

For the in vitro labeling of control (WT/*2D2*) and kindlin-3-deficient (*Fermt3<sup>-/-</sup>/CD4-Cre/2D2*) CD4[+] T cells (*Moretti et al., 2013*), CellTrace CFSE Cell Proliferation Kit (Thermo Fisher) was used to trace dividing cells. Briefly, single cell suspensions from spleens were treated with ammonium-chloridepotassium for RBC lysis and CD4 T cells were sorted using CD4 T cell isolation kit (Milteniy Biotech). T cells were then stained with CFSE according to the manufacturer's instructions and about $5 \times 10^4$ T cells per sample were cocultured in triplicate in 96-well round bottom plate with $2 \times 10^4$ DCs and increasing concentration of soluble MOG35-55 peptide (0.1, 1 and 10 µg/mL). For T cell stimulation with anti-CD3e and anti-CD28 Abs, $2 \times 10^5$ CFSE-labelled cells per sample were cultured in wells pre-coated for 1 hr at 37°C with anti-CD3e (10 µg/ml, eBioscience), with soluble anti-CD28 (2 µg/ml, eBioscience) and PMA (20 ng/ml). Proliferation was analysed by FACS after 3 days of incubation in R10 (RPMI-1640, 10% FCS, 1x Penicillin-Streptomycin) at 37°C.

## Time-lapse visualization of thymus attraction

Time-lapse imaging of horizontal cell migration to fetal thymus lobes was recorded as previously described (*Liu et al., 2005*; *Ueno et al., 2005*). Briefly, 10 µL of freshly neutralized collagen solution (1.72 mg/mL PureCol, Advanced BioMatrix, in RPMI 1640-based culture medium) containing 1–2 × $10^5$ FL-Lin<sup>neg</sup> cells from either *Fermt3<sup>+/+</sup>* or *Fermt3<sup>-/-</sup>* C57BL/6 (CD45.2[+]) animals was placed in a 35 mm plastic dish and solidified at 37°C for 10 min. An E15.5 fetal thymus lobe from wild-type SJL (CD45.1[+]) mouse, pre-treated for 6 days with 1.35 mM 2-deoxyguanosine (dGuo; Sigma) was positioned approximately 0.5 mm away from the cell spot. The culture was submerged in 1.72 mg/mL collagen medium and solidified at 37°C for 30 min. The dish was placed in 5% $CO_2$ under an Axiovert 40 CFL microscope (Zeiss) equipped with a digital CCD camera. The culture was time-lapse

monitored for 36 to 48 hr. After colonization, fetal thymus lobes were removed from the collagen gel, rinsed, and further cultured for 18 days under conventional organ culture conditions (see below). Thymi were than pooled, squeezed through a cell strainer and the resulting single cell suspensions were stained with anti-CD45.2 to identify FL cell-derived thymocytes and further gated for anti-CD4-FITC and anti-CD8-APC.

## Fetal Thymus Organ Culture

Seeded thymus lobes were cultured on 0.8 μm isopore membrane filters (Millipore) supported by a Gelfoam sponge (Pfizer) at an interface between 5% $CO_2$-humidified air and RPMI 1640-based culture medium containing 10% FCS, 100 U/ml penicillin and 100 μg/ml streptomycin (PAA), 2 mM L-glutamine (PAA), 1X non-essential amino acids (Gibco), 1 mM sodium pyruvate (Gibco), 10 mM HEPES, 50 μM 2-mercaptoethanol (Sigma). Details have been described (*Ueno et al., 2005*).

## Blood flow velocity measurement

Blood flow velocity measurements were performed in the fetal yolk-sac vasculature of E12.5 to E16.5 fetuses as described (*Margraf et al., 2017*). Briefly, pregnant WT mice were anesthetised and fetuses surrounded by the yolk sac and still connected to the placenta were surgically exteriorized and placed on a modified intravital microscopy stage. Manually prepared glass microcapillaries (Clark Capillaries GCI150TF-10, Clark Electromedical Instruments, Pangbourne Reading) were used to inject 4 μl of a 5% FITC dextrane solution (FD150S, Sigma-Aldrich, Taufkirchen, Germany) with 1 μl of red fluorescent microspheres (FluoSpheres polystyrene microspheres 1,0 μm red fluorescent 580/605; Invitrogen) into the fetal vasculature. Imaging was performed using an in vivo microscope setup (Olympus BX1, Olympus, Hamburg, Germany) equipped with appropriate filter sets and a double-flash device (Rapp Optoelectronics, Hamburg, Germany). Images were recorded using a 60x objective (Olympus LUM Plan FI/IR 60x/0.90W) with setup specific CCD-cameras (Kappa CF5 HS; Kappa Optronics GmbH, Gleichen, Germany, and LaVision Imager ProX pco.1600L, LaVision GmbH, Goettingen, Germany) and recorded utilizing VirtualDub (www.virtualdub.org) and Imspector Pro Software (Lavision BioTec GmbH, Bielefeld, Germany). Analysis was performed using ImageJ (National Institute of Health). Vessel diameters were chosen between 20 and 50 μm.

## Flow chamber assays

Flow chamber assays were performed with the ibidi pump system (Martinsried, Germany). Ibidi flow chambers (μ-Slide I 0.8 Luer ibiTreat) were coated with 10 μg/ml recombinant mouse (rm) ICAM-1-Fc or rmVCAM-1-Fc, 10 μg/ml rmP-selectin-Fc, 5 μg/ml rmCCL21 and 5 μg/ml rmCCL25 (all from R & D systems) over night at 4°C. After blocking for 1 hr with 1% casein (Thermo Fisher Scientific) at room temperature, the channels were filled with control or $Fermt3^{-/-}$ $Pir^+$ T cell progenitors sorted from E13.5 fetal livers. After 3 min incubation without flow, slow flow of approximately 0.15 dyn/cm$^2$ was applied for 3 min followed by increasing flow of 1 and 3 dyn/cm$^2$ for 10 min each. Pictures were taken every 5 s for a total of 25 min at an Axiovert 40 CFL microscope (Zeiss) equipped with a digital CCD camera and a 10 x objective. The number of cells that were adherent immediately before the start of the fast flow was set to 100%. Cells that did not change position within 10 s were considered to be firmly adherent.

## Live cell imaging of T cell adhesion within the lymph node vasculature and flow in vivo

CD4 T cells were isolated from spleen by negative selection as described above. Anaesthetized mice were injected intravenously with a 1:1 mixture of CFSE and Far Red labelled WT and *Fermt3* hypomorphic CD4 T cells. The hair of the hind legs was removed and mice were placed into an imaging chamber with integrated heating plate. In a dorsal approach the skin of the hind legs was cut and the popliteal lymph nodes were exposed. To reveal the superficial vessels of the lymph nodes, they were fixed to the biceps femoris muscle using Histoacryl tissue adhesive (B. Braun). PBS-soaked batting was placed around the legs to keep the tissue moist. The lymph nodes were covered with a cover glass, which was topped with PBS to facilitate imaging. Time-lapse video microscopy was performed using an upright spinning disk confocal microscope (Examiner, Zeiss, Germany) equipped with a confocal scanner unit CSU-X1 (Yokogawa Electric Corporation, Japan), an EMCCD camera

(Evolve, Photometrics, USA) and a 20x/1.0NA water immersion objective (Zeiss, Germany). 4D images (30 z-stacks with a step size of 4 µm and an average time lapse interval of 42 s) were acquired by using two lasers with an excitation wavelength of 488 nm and 640 nm. Adhesion was analyzed by generating sum projections of z-stacks over time using Slidebook 6.0.8 Software (3i, USA). The number of adherent leukocytes was determined as cells attached to the same position for more than 1 min using Fiji/ImageJ software (NIH, USA). Centerline blood flow velocities in the same vessels were obtained upon subsequent intravenous injection of $10^7$ fluorescent microspheres (1 µm in diameter; FluoSpheres 580/605, Invitrogen) by measuring frame-to-frame displacement of single beads in LN microvessel segments. Images of one plane with an average time lapse interval of 66.5 ms were acquired for one minute using one laser with an excitation wavelength of 561 nm. Wall shear rates were calculated as previously described (*Sperandio et al., 2006*).

## Statistic analysis

Means, standard deviations and statistical comparisons with Student's t-test (*$p<0.05$, **$p<0.01$, ***$p<0.001$) were made using Microsoft Excel software.

## Ackowledgments

We thank Arnoud Sonnenberg for critically reading the manuscript and the Fässler lab members for discussion, Michal Grzejszczyk and Kerstin Kraus for excellent technical assistance and Eric Vroegin-deweij for providing the anti-ER-TR4 and 5 antibodies. The work was supported by the Deutsche Forschungsgemeinschaft (SFB914; Projects A01, A05, B01, B09 and Z03) and the Max Planck Society.

## Additional information

### Competing interests

Reinhard Fässler: Reviewing editor, *eLife*. The other authors declare that no competing interests exist.

### Funding

| Funder | Grant reference number | Author |
|---|---|---|
| Deutsche Forschungsge-meinschaft | SFB914 | Federico Andrea Moretti<br>Sarah Klapproth<br>Raphael Ruppert<br>Andreas Margraf<br>Jasmin Weber<br>Robert Pick<br>Christoph Scheiermann<br>Markus Sperandio<br>Reinhard Fässler<br>Markus Moser |
| Max-Planck-Gesellschaft | Open-access funding | Markus Moser<br>Reinhard Fässler |

The funders had no role in study design, data collection and interpretation, or the decision to submit the work for publication.

### Author contributions

Federico Andrea Moretti, Investigation, Writing—original draft, Carried out most experiments and data analyses, Evaluated and interpreted the data; Sarah Klapproth, Investigation, Carried out most experiments and data analyses, Evaluated and interpreted the data; Raphael Ruppert, Investigation, Generated and analysed chimeric mice; Andreas Margraf, Investigation, Measured blood flow veloci-ties and shear rates in the yolk sac vasculature; Jasmin Weber, Robert Pick, Investigation, Performed and analysed the lymph node homing experiments; Christoph Scheiermann, Supervision, Investiga-tion, Supervised the lymph node homing experiments and measured integrin ligand expression in

the vasculature; Markus Sperandio, Supervision, Supervised the blood flow and shear rate measurements in the yolk sac vasculature; Reinhard Fässler, Supervision, Funding acquisition, Writing—review and editing; Markus Moser, Supervision, Funding acquisition, Investigation, Writing—original draft, Writing—review and editing

### Author ORCIDs
Federico Andrea Moretti https://orcid.org/0000-0003-3187-3683
Markus Sperandio https://orcid.org/0000-0002-7689-3613
Markus Moser https://orcid.org/0000-0001-8825-5566

### Ethics
Animal experimentation: Animal experimentation: Housing and use of laboratory animals at the Max Planck Institute of Biochemistry are fully compliant with all German (e.g. German Animal Welfare Act) and EU (e.g. Annex III of Directive 2010/63/EU on the protection of animals used for scientific purposes) applicable laws and regulations concerning care and use of laboratory animals. All of the animals were handled according to approved licenses from the Government of Upper Bavaria (55.2-1-54-2532-96-2015 and 55.2-1-54-2532-175-2011)

### Decision letter and Author response
Decision letter https://doi.org/10.7554/eLife.35816.021
Author response https://doi.org/10.7554/eLife.35816.022

## Additional files

### Supplementary files
• Transparent reporting form
DOI: https://doi.org/10.7554/eLife.35816.018

### Data availability
All data generated or analysed during this study are included in the manuscript and supporting files.

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
