## [Decision Letter]

Thank you for submitting your article "Differential requirement of kindlin-3 for T cell progenitor homing to the non-vascularized and vascularized thymus" for consideration by *eLife*. Your article has been reviewed by three peer reviewers, including Holger Gerhardt as the Reviewing Editor and Reviewer #1, and the evaluation has been overseen by Randy Schekman as the Senior Editor. The following individual involved in review of your submission has agreed to reveal their identity: Andrés Hidalgo (Reviewer #2).

The reviewers have discussed the reviews with one another and the Reviewing Editor has drafted this decision to help you prepare a revised submission.

Summary:

Moretti and colleagues reinvestigate the kindlin-3 knockout mice to study the importance of integrin mediated adhesion during T-cell homing to the thymus. They find that early embryonic homing of fetal liver derived progenitor cells proceeds in the absence of strong integrin adhesion, but post-natal thymus atrophy ensues due to defective homing of bone marrow derived circulating progenitor cells to the vascularised thymus. The authors perform a logical series of experiments to understand the mechanistic basis for the differences in kindlin-3 requirement, and come to the conclusion that increased shear forces present at the later developmental stages interfere with homing of weakly adhesive T-cells.

Overall this is a carefully executed and well documented study that provides valuable insights into in vivo mechanisms at the interface of mechanobiology and organogenesis. The main finding of a selective defect in kindlin-3 deficient T-cells and their homing mechanism when cardiovascular development has proceeded to drive larger hemodynamic shear forces is considered well worth reporting to a wider scientific audience.

Having said that, the reviewers agree this main conclusion should be strengthened in two key areas:

Essential revisions:

The referees request that you:

1) Provide a more direct proof of shear rates in thymus or more generally present proof that adhesion of WT vs. kindlin-3 mutants is differentially affected at varying flow/shear values. Is it possible to rescue the defect by lowering shear rates in the thymus, or potentially study T-cell behaviour in organs with different shear rates? Alternatively, you may wish to expand the intravital imaging approach to characterise T-cell adhesion differences under different hemodynamic shear in vivo.

2) Show that the thymic defects are hematopoietic-derived (i.e. not cause by expression in other cell types). Here they suggest either conditional deletion of kindlin-3, or more practically transferring null T-cells to Wt animals.

---

## [Author Response]

Essential revisions:The referees request that you:1) Provide a more direct proof of shear rates in thymus or more generally present proof that adhesion of WT vs. kindlin-3 mutants is differentially affected at varying flow/shear values. Is it possible to rescue the defect by lowering shear rates in the thymus, or potentially study T-cell behaviour in organs with different shear rates? Alternatively, you may wish to expand the intravital imaging approach to characterise T-cell adhesion differences under different hemodynamic shear in vivo.

We agree with the reviewers that this is a central point of our study. However, in our opinion a direct measurement of shear rates within the developing thymus or alternatively lowering the shear rates within the thymus is technically not feasible also because the mutant mice die shortly after birth.

We tried to address this point by several other means. Firstly, we transferred a 1:1 mixture of differently labelled CD4 T cells from spleens of kindlin-3-floxed/CD4-Cre positive and negative (control) mice (in total 6x10^6^ cells) via a catheter of the carotid artery into wild-type mice. 6 and 2 h before preparation of the cremaster muscle mice were treated by intrascrotal injection of IFNγ and TNFα, respectively, to induce local inflammation and T cell adhesion. We chose the cremaster muscle, because here we easily find vessels with different flow rates. However, we hardly recorded fluorescent cells passing through cremaster muscle venules and only detected very few adherent cells. Thus, unfortunately we were unable to correlate flow rates with the adhesion characteristics of wild-type vs. kindlin-3^-/-^ T cells.

Secondly, since kindlin-3 regulates many, if not all integrin classes expressed on hematopoietic cells ranging from HSCs, platelets, myeloid to lymphoid cells in a similar fashion, we analysed adhesion of wild-type and kindlin-3 mutant PMNs to cremaster muscle venules. These datasets also included the blood flow velocities of the individual vessels and therefore allowed a correlation of the adhesion efficiencies versus shear rates. As shown in Author response image 1 we found despite a high variance a slight increase in adhesion efficiency at higher shear rates for wild-type PMNs, while adhesion efficiencies rather decreased in kindlin-3^-/-^ PMNs at higher shear rates. Since adhesion of kindlin-3^-/-^ PMNs is already very low at low shear conditions, we also included measurements on kindlin-3 hypomorphic mice into our analysis, which express 5% of kindlin-3 (Klapproth et al., 2015). These cells show a higher basic adhesion but like kindlin-3^-/-^PMNs decreased adhesion at higher shear rates (see Author response image 1).

**Author response image 1. respfig1:** Adhesion efficiencies of PMNs in cremaster muscle venules of wild-type (WT), Kindlin-3 hypomorph (neo) mice and kindlin-3 knockout (KO) chimeras at indicated shear rates.

In a third experimental approach we imaged T cell adhesion to the lymph node vasculature by intravital microscopy. Since kindlin-3 deficient cells hardly adhere to the vascular wall, we adoptively transferred CD4 T cells from wild-type and kindlin-3 hypomorphic mice. Both T cell populations were labelled differently with fluorescent dyes and injected in a 1:1 ratio into wild-type mice (15x 10^6^ cells/mouse). Consequently, the distribution of adherent wild-type and kindlin-3 hypomorphic T cells in lymph node vessels with different flow rates was recorded. These experiments showed that wild-type T cells primarily adhered to blood vessels with higher shear rates, whereas kindlin-3 hypomorphic T cells were found in areas with lower blood flow velocities and shear rates. These data are now included in the manuscript in Figure 8G-K.

2) Show that the thymic defects are hematopoietic-derived (i.e. not cause by expression in other cell types). Here they suggest either conditional deletion of kindlin-3, or more practically transferring null T-cells to Wt animals.

To address this question we have performed three sets of experiments. First, we have generated mixed fetal liver cell chimeras, in which we transferred wild-type or kindlin-3^-/-^ FL cells (CD45.2) together with wild-type FL cells from congenic B6.SJL mice (CD45.1). Since we have previously shown that kindlin-3 deficient HSCs have a 5-fold reduced homing capacity to the bone marrow (Ruppert et al., 2015), we circumvent the homing defect be transferring 9x more CD45.2 FL cells than FL cells from B6.SJL mice (9:1 chimeras). Chimeras, which received wild-type CD45.2 cells showed a perfect 9:1 ratio in all thymic populations (DN, DP, CD4 and CD8). In contrast, 9:1 kindlin-3^-/-^ chimeras showed a strong reduction in the DN population, clearly demonstrating that kindlin-3^-/-^T cell progenitors have a strong disadvantage compared to wild-type cells in homing to the thymus. These data are depicted in Figure 4—figure supplement 1B.

Second, we conditionally deleted kindlin-3 in the hematopoietic system of kindlin-3-floxed/Mx1-Cre mice. Three weeks after polyIC injection the DN T cell populations from thymi of Cre-positive and Cre-negative kindlin-3-floxed mice were analysed by flow cytometry. These experiments revealed a complete absence of DN cells in conditional kindlin-3^-/-^mice indicating that the T cell progenitor homing defect to the thymus is hematopoietically-derived (see Figure 4—figure supplement 1A).

Third, in order to directly show that the thymic defects are based on a hematopoietic defect and not caused because kindlin-3 is also expressed in a non-hematopoietic thymic cell type such as epithelial cells, we performed flow cytometric experiments with kindlin-3-EGFP knockin mice, in which we tagged Kindlin-3 C-terminally with EGFP. The mice express normal kindlin-3-EGFP fusion protein levels (unpublished). These experiments show that only CD45 positive cells express EGFP in the thymus. CD31 pos endothelial cells and EpCAM pos thymic epithelial cells are EGFP negative. We show these important data in Figure 4—figure supplement 2.